# Comparing the safety and efficacy of sodium valproate, levetiracetam, and phenytoin in attenuating the severity of agitation in patients with post-traumatic brain injury: An observational study

Shivani Singh[1], Raghavendra Nayak[2], Shivaprakash Gangachannaiah [1]*,
Ashutosh Bhosale[3], Reena Sherin Parveen[1], Varun Kumar S. G. [4], Geeta Sundar[2]

1 Department of Pharmacology, Kasturba Medical College, Manipal Academy of Higher Education,
Manipal, Karnataka, India, 2 Department of Neurosurgery, Kasturba Medical College, Manipal Academy
of Higher Education, Manipal, Karnataka, India, 3 Department of Pharmacy Practice, Manipal College
of Pharmaceutical Sciences, Manipal Academy of Higher Education, Manipal, Karnataka, IIndia,
4 Department of Applied Statistics and Data Science, Prasanna School of Public Health, Manipal
Academy of Higher Education, Manipal, Karnataka, IIndia

* shiva.g@manipal.edu

## Abstract

### Objective

The present study aimed to compare the safety and efficacy of sodium valproate,
levetiracetam, and phenytoin in agitation control.

### Methods and Material

This prospective observational study included 189 adult patients with traumatic
brain injury (TBI) receiving sodium valproate, phenytoin, and levetiracetam. Agitation
was monitored using the Richmond Agitation-Sedation Scale (RASS) at baseline
and serially over seven days. The study evaluated the percentage of patients who
experienced agitation-relief (A-R) following antiepileptic drugs, along with pattern of
adverse events among the three groups.

### Results

Sodium valproate (sod.v) demonstrated better efficacy with 85.7% of patients achieving A-R compared to 63.5% in levetiracetam (P = 0.016). The median time to achieve
A-R was 3 days in sod.v and significantly lesser compared to 4 days in phenytoin
(P = 0.04) and 5 days in levetiracetam (P = 0.0001). By day 4, 72% of patients in the
sod.v achieved A-R, in contrast to 50% in the phenytoin and 31.7% in the levetiracetam group. The safety profile of sod.v and levetiracetam was more favorable with
lesser occurrence of adverse events compared to phenytoin (P = 0.017).

e0350585. https://doi.org/10.1371/journal.
pone.0350585

and Reggio Emilia, ITALY

**Peer Review History:** PLOS recognizes the
benefits of transparency in the peer review
process; therefore, we enable the publication
of all of the content of peer review and
author responses alongside final, published
articles. The editorial history of this article is
available here: https://doi.org/10.1371/journal.
pone.0350585

**Data availability statement:** We confirm that the submission contains all data required to replicate the study findings, including the values underlying summary statistics, figures, and analyses, which are provided as Supporting Information. However, since explicit consent for unrestricted public data sharing was not secured at the time of enrolment from the participants, full open access to the dataset is restricted to ensure participant confidentiality. The underlying data are available upon request through the Institutional Ethics Committee (IEC), Kasturba Medical College, Manipal. Requests for data access may be directed to: iec.kmc@manipal.edu to ensure compliance with ethical and legal requirements.

**Funding:** The author(s) received no specific funding for this work.

**Competing interests:** The authors have declared that no competing interests exist.

## Conclusions

To our knowledge, this is the first pivotal evidence to compare antiepileptics for agitation control in post-traumatic brain injury patients. Our study demonstrated that patients receiving sodium valproate showed relatively greater and earlier improvement in agitation control, with an acceptable safety profile.

The study proves the dual benefits of sod.v in post-TBI improving patient outcomes and alleviating mental strain on patients, and their families.

## Introduction

Traumatic brain injury (TBI) is a significant concern globally. It is a major cause of death and serious disability among all age groups, causing a disproportionate burden on treatment and the economic front [1,2]. Post-traumatic agitation is a state of confusion that occurs following the initial injury, marked by emotional unrest, akathisia, impulsivity, disinhibition, aggression, and a diminished capacity to sustain or suitably change attention [3,4].

Agitation and delirium are common neuropsychiatric complications in patients with traumatic brain injury (TBI), particularly in neuro-intensive care settings. Delirium is highly prevalent in Neuro-Intensive Care Units (NICUs), affecting 12–43% of neurocritical care patients and up to 87% of elderly ICU populations [5–7]. Among elderly trauma patients, delirium incidence ranges from 10–40%, while prevalence in ICU patients with traumatic brain injury (TBI) may reach 60% [8–11]. Agitation poses immediate risks of injury to patients, caregivers, and healthcare staff, disrupts functional recovery, and increases the demand for monitoring and intervention. Untreated agitation contributes to extended hospital stays, higher healthcare costs, and poor reintegration into the community.

Delirium arises from multifactorial biological mechanisms, including neuroinflammation and stress responses [12,13] neurotransmitter imbalance [13,14] cerebral hypoperfusion and hypoxia [15], and structural and functional brain alterations [16,17]. Biomarker evidence supports neuronal injury involvement: S100β reflects astrocytic injury and blood–brain barrier disruption, whereas neurofilament light chain (NfL) indicates axonal damage associated with delirium duration and cognitive outcomes. GFAP and UCH-L1 show diagnostic utility in TBI, while elevated S100β and matrix metalloproteinases in subarachnoid haemorrhage correlate with blood–brain barrier disruption and delayed cerebral ischaemia. Multimodal biomarker integration may enable biological delirium subtyping and precision approaches [17].

Effective management of agitation is essential to improve patient outcomes and optimize healthcare efficiency. Many medications, including antipsychotics (haloperidol, quetiapine, and olanzapine) [4,18], anxiolytics (buspirone and benzodiazepines) [19,20], beta-blockers (propranolol) [21–23], antidepressants (fluoxetine, paroxetine, and amitriptyline) [3,4,22,24] and lithium [24], are used to manage agitation. However, these medications are associated with notable adverse effects, such as QTc interval prolongation, restlessness, dystonic reactions, hypotension, and bradycardia.

Currently, management emphasizes prevention through multicomponent non-pharmacological strategies (early mobilisation, sleep hygiene, reorientation, sensory aids) and optimisation of modifiable risk factors (avoidance of deliriogenic drugs, correction of pain, hydration, and electrolyte imbalances) [9,10,12,13,25]. These are supplemented in ICU settings by breathing techniques, and judicious sedation [13]. While specialised delirium care models may enhance results, pharmacological therapy (e.g., haloperidol) is reserved for severe agitation and administered judiciously [12,13,26,27].

Although AEDs are widely used in patients with traumatic brain injury (TBI) for seizure prevention and control, there remains considerable uncertainty regarding the optimal agent and duration of therapy [28]. Current clinical practice varies substantially, reflecting the limited comparative evidence available. Ongoing randomized studies aim to address this gap, most notably the MAST Trial (Pharmacological Management of Seizures Post Traumatic Brain Injury; ClinicalTrials.gov Identifier: NCT04573803) [29], a large multicentre phase III study designed to evaluate both the duration of AED therapy after early post-traumatic seizures and the effectiveness of prophylactic treatment with phenytoin or levetiracetam in severe TBI. The results of this trial are expected to clarify best practice in AED management following TBI. Currently, management emphasizes prevention through multicomponent non-pharmacological strategies including early mobilization, sleep hygiene, reorientation, and sensory aids [9,10,12,13,30], risk-factor optimization such as avoidance of deliriogenic medications (e.g., benzodiazepines, anticholinergics), management of pain, hydration, electrolyte imbalances [9,10,13]. In ICU settings, this is supported by interventions such as judicious sedation, ventilation, and physical therapy. Antipsychotics such as haloperidol are reserved for severe agitation and require cautious use [12,13,25,26] while specialized delirium care models may improve outcomes.

Valproic acid (VPA) has been investigated as a therapeutic option for agitation and behavioural dysregulation following traumatic brain injury (TBI), although clinical findings remain heterogeneous. A retrospective chart review demonstrated improvement in agitation symptoms with VPA administered at doses comparable to conventional psychiatric practice (approximately 1250 mg/day) Evidence from randomized trial reported no significant effects of VPA on neuropsychological functioning after TBI. Despite variable efficacy outcomes, VPA offers practical advantages, including a lower propensity for sedation, irritability and reduced cognitive impairment compared with alternative agents, potentially facilitating greater participation in rehabilitation and ongoing neurological assessment [27,31]. Sodium valproate (sod.v), phenytoin, and levetiracetam are among the commonly used anticonvulsants. Many scientific studies have shown that sod.v is particularly effective in alleviating the severity of agitation [31–33]. Since there is an absence of standardized treatment protocols specifically tailored to agitation control and given the scarcity of information about the role of anticonvulsants in attenuating agitation in post-TBI patients, the purpose of the current study was to evaluate and compare the efficacy of sod.v, levetiracetam and phenytoin in controlling agitation in post-traumatic brain injury patients.

## Subjects and methods

The Institutional Ethics Committee (IEC) approved the study protocol under reference number IEC2:407/2022. Data collection for the first patient commenced following registration with the Clinical Trials Registry-India (CTRI): CTRI/2023/01/049333. Recruitment period was from 1st Jan 2023–31 Dec 2024. A patient information sheet (PIS) in English or the local language was provided, and a written informed consent was obtained. Approval was obtained from a legally authorized representative for those unable to consent.

### Study design and population

The present study was a hospital-based, non-randomized prospective observational study conducted over 19 months between February 2023 to July 2024 in the Department of Neurosurgery at a tertiary care hospital. All new ICU admissions were screened for study eligibility and eligible patients receiving sodium valproate, phenytoin, or levetiracetam were approached for informed consent. Participants enrolled from multiple clinical units and the treating neurosurgeons made all the decisions about the choice of AEDs. The commonly used AEDs were either sod.v, phenytoin, or levetiracetam as antiepileptic

drugs for seizure prophylaxis. In order to achieve a balanced representation of each treatment group, participants were enrolled sequentially in 1:1:1 ratio corresponding to the three drugs used across the units. This approach ensured that each group received an equal number of patients, with a total of 63 patients in each group by the conclusion of enrollment.

Inclusion criteria required all post-TBI cases in the age group of 18–65 years who were willing to get admitted and give informed consent by a legally authorized representative if incapacitated. Patients who had RASS score of more than equal to one were included. Patients requiring surgical interventions, history of epilepsy, neurodegenerative diseases (e.g., Alzheimer's, Parkinson's), or psychotic disorders, patients taking antipsychotic medications or other antiepileptic drugs, hepatic or renal disease, history of substance abuse, alcohol use, or pregnant and lactating women were excluded.

## Study variables

Baseline demographic data of all the patients including age, sex, comorbidity, drugs, cause of injury, blood pressure, heart rate, oxygen saturation, Marshall score, and Glasgow Coma Scale score, were assessed, monitored, and recorded. The Richmond agitation sedation score (RASS) is a tool employed for the assessment of agitation and scoring. It is a validated tool for assessing alertness and agitation in ICU patients. It helps guide sedation management, provides clear criteria for evaluating arousal and agitation, and facilitates communication among healthcare providers. The RASS is a ten-point scale, with scores ranging from −5 (deep sedation or unarousable) to +4 (severe agitation or combative behavior), and a score of 0 indicates an "alert and calm" state. Scores are based on the patient's responses to auditory and physical stimuli, ensuring consistent and objective assessment. The Marshall grading system categorizes TBI based on CT scan findings. In Category I, there is no visible intracranial pathology. Category II involves a midline shift of 0–5 mm with visible basal cisterns and no high or mixed-density lesions larger than 25 cm$^3$. Category III also has a midline shift of 0–5 mm but with compressed or absent basal cisterns, while the lesions remain under 25 cm$^3$. In Category IV, a midline shifts greater than 5 mm is present without large high or mixed-density lesions. Category V involves any lesion that has been surgically evacuated, while Category VI includes high or mixed-density lesions larger than 25 cm$^3$ that have not been surgically treated [34,35]. For adverse drug reactions, the ADR FORM VERSION 1.4 from Central Drugs Standard Control Organization (CDSCO) was used, and for causality assessment, the Naranjo Adverse Drug Reaction Probability Scale was employed. Post-traumatic brain injury patients received antiepileptic drugs as a prophylaxis for seizures. Sod.v was administered at a dose of 400 mg twice daily (BD), phenytoin at 100 mg three times daily (BD), and levetiracetam at 1000 mg two times daily (BD).

All patients received the two first doses intravenously. However, 14.2% of patients in sod.v, 12.6% in phenytoin, and 17.4% in levetiracetam needed to continue receiving IV medication after 24 hours of admission, according to their reaction and the neurosurgeon's evaluation. The participants' enrolment flowchart is depicted in Fig 1.

## Sample size and sampling procedure

Using G∗ power software, the sample size was computed. With an effect size of 0.25, assuming a significance level (α) of 0.05, statistical power of 0.8, the calculated sample size was 156, with approximately 52 subjects per group. Accounting for the anticipated attrition rate of 15–20% (observed from the previous hospital records), the sample size was adjusted to 63 participants per group, and a total of 189 participants was finally decided for the study. Data on RASS, Marshall score, BP, saturation, and GCS, were collected over seven days according to the data collection protocol following admission for post-TBI care. If a patient was discharged before the seventh day, observations were recorded up to the day of discharge. Data on any additional drugs prescribed to the patient during the study period and adverse effects due to drugs were recorded.

## Outcome measures

The primary outcome measure was the percentage of patients achieving agitation-relief (A-R) and the time to achieve it among the three groups. A-R was defined as the number of patients attaining a RASS score of zero following antiepileptic

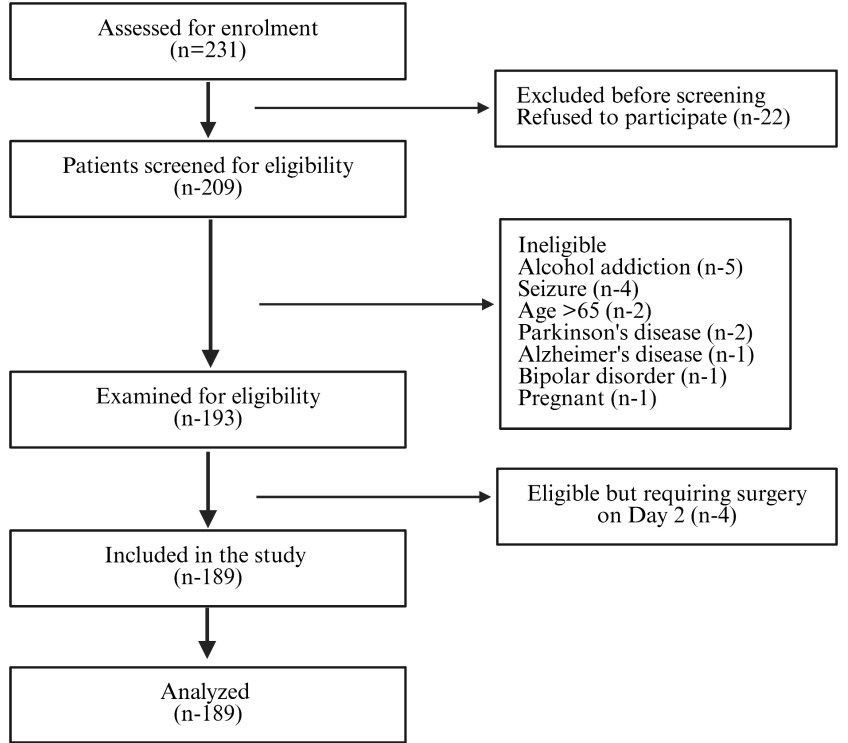

**Fig 1. STROBE Flow Diagram Depicting the participant enrollment in the study.** n = Number of participants.

drug therapy. The secondary outcome measures included the frequency and types of adverse events among the study groups.

## Statistical analysis

Microsoft Excel was used for data compilation and analyses were performed using R software (Version 4.4.1). The Shapiro-Wilk test was used to assess the normality of the data set. Categorical variables were presented as percentages and continuous variables as median and inter-quartile range (IQR). The chi-square test was used to assess the association between the study groups, and the Kruskal-Wallis test was used to assess the relation between the continuous variables. The Kaplan-Meier method was used to estimate the time to achieve agitation-relief among the three groups and Peto–Peto–Wilcoxon test was applied to compare the time-to-event distributions between the groups. A p-value of less than 0.05 was considered statistically significant.

## Results

The baseline characteristics were similar across all the groups studied. Sex distribution and vital parameters, including systolic blood pressure, diastolic blood pressure, heart rate, and oxygen saturation (SPO2), were not significantly different between the groups. There were no significant differences observed in comorbidities, baseline agitation scores, GCS scores, or type of head injury. Patients with Marshall score grade I restraint was significantly more common in the phenytoin group than in the levetiracetam group as shown in Table 1.

**Table 1. Baseline characteristics of patients.**

| Variables | Sodium Valproate (N = 63) | Phenytoin (N = 63) | Levetiracetam (N = 63) | P value |
|---|---|---|---|---|
| Age(in years) | 41.00(28-50) | 37.00(25-51) | 33.0(28-52) | 0.693 |
| `Gender n(%) | | | | 0.507 |
| Male | 51(81.0) | 47(74.6) | 52(82.5) | |
| Female | 12(19.0) | 16(25.4) | 11(17.5) | |
| SBP (mmHg) | 130(118-132) | 130(120-140) | 130(120-140) | 0.249 |
| DBP (mmHg) | 80(70-80) | 80(76-90) | 80(70-80) | 0.013* |
| HR(Bpm) | 86(76-94) | 88(80-96) | 84(74-92) | 0.259 |
| SPO2(%) | 99(98-99) | 99(98-100) | 99(98-100) | 0.494 |
| Comorbidities n (%) | | | | 0.673 |
| Hypertension | 6(46.1) | 7(53.8) | 12(70.6) | |
| Diabetes | 3(23.1) | 3(23.1) | 1(5.9) | |
| Both | 4(30.8) | 3(23.1) | 4(23.5) | |
| RASS n (%) | | | | 0.083 |
| RASS < 1 | 12(19.1) | 20(31.7) | 23(36.5) | |
| RASS ≥ 1 | 51(80.9) | 43(68.3) | 40(63.5) | |
| GCS | 13(11-14) | 13(12-14) | 12(10-13) | 0.057 |
| Severity of injury n (%) | | | | 0.083 |
| Mild | 46(73.0) | 47(74.6) | 34(53.9) | |
| Moderate | 10(15.9) | 12(19.1) | 18(28.6) | |
| Severe | 7(11.1) | 4(6.3) | 11(17.5) | |
| Marshall score n (%) | | | | 0.026* |
| I | 7(11.1) | 14(22.2)# | 3(4.8) | |
| II | 46(73.1) | 40(63.5) | 48(76.2) | |
| III | 6(9.5) | 2(3.2) | 2(3.2) | |
| IV | 4(6.3) | 7(11.1) | 10(15.8) | |
| Cause of injury n(%) | | | | 0.325 |
| Road traffic accident | 52(82.5) | 51(81) | 50(79.4) | |
| Blunt trauma | 9(14.3) | 7(11.1) | 12(19) | |
| Others | 2(3.2) | 5(7.9) | 1(1.6) | |

Continuous data presented as medians and interquartile ranges in parenthesis. Categorical data presented as numbers and percentages in parentheses. *p ≤ 0.05 was considered statistically significant. "GCS- Glasgow Coma Scale, SBP-systolic blood pressure, DBP-diastolic blood pressure, SpO2-oxygen saturation, RASS-Richmond agitation sedation scale." #p = 0.004 for Marshall score I discordance between phenytoin and levetiracetam. Chi-square test used to assess the association between the study groups, and Kruskal-Wallis test was used to assess the relation between the continuous variables.

There was a significant increase in the percentage of patients who achieved A-R following sod.v (85.7%) compared with levetiracetam (63.5%) (P = 0.004). There was no significant association between phenytoin and the other two groups as shown in Table 2.

A significantly higher proportion of patients in the sod.v group achieved A-R when compared to the other 2 groups (P = 0.0001). Furthermore, the graph demonstrates that the sod.v group achieves A-R more rapidly than the other two drugs. As illustrated in Fig 2, at the end of day 4, nearly 72% (18 of 63) of the head injury patients had experienced A-R in the sod.v group, whereas in the phenytoin and levetiracetam groups, only 50% and 31.7%, respectively, experienced A-R.

**Table 2. Agitation-relief among head injury patients treated with prophylactic antiepileptic drugs.**

| Groups | n | Agitation-relief n (%) | Agitation not relieved n (%) | P value |
|---|---|---|---|---|
| Sod. Valproate | 63 | 54(85.7) * | 9(14.3) | 0.0166 |
| Phenytoin | 63 | 47(74.6) | 16(25.4) | |
| Levetiracetam | 63 | 40(63.5) | 23(36.5) | |

Chi-square test was applied. *P=0.004 compared to levetiracetam (significance by Bonferroni corrected *p*-value).

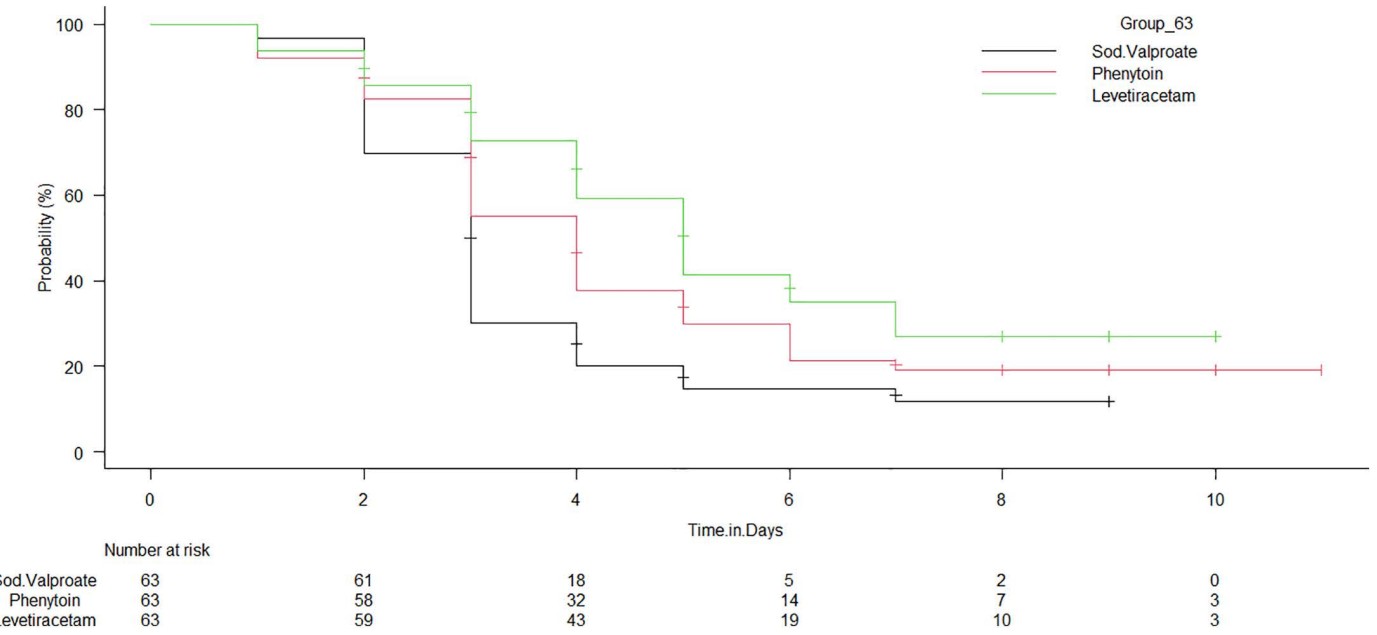

**Fig 2. Kaplan–Meier estimates of time to Agitation-Relief among treatment groups.** The time to achieve agitation-relief in days is represented on the X-axis, and the probability of achieving A-R is represented on the Y-axis. The Peto–Peto–Wilcoxon test was applied to compare the curves (P=0.005).

Patients in the sodium valproate group achiedved earlier improvements of agitation within the initial days of treatment as shown in Table 3. The median time to achieve A-R was less for sod.v than for levetiracetam and phenytoin (P=0.0001).

Adverse drug reactions were reported significantly higher in the phenytoin group compared to sod.v and levetiracetam (p=0.03). The occurrence of vertigo was more in phenytoin compared to sod.v and levetiracetam groups (P=0.003). Hyponatremia was observed more in levetiracetam compared to phenytoin(P=0.01). Hypersensitivity reaction was observed in phenytoin group (Fig 3). No liver or renal function abnormalities were identified within the duration of the study period. Based on the Naranjo Adverse Drug Reaction (ADR) assessment scale, the causality of all reported adverse reactions was determined as possible.

## Discussion

Traumatic brain injury patients were administered antiepileptics for early post-traumatic seizure prophylaxis (PTS) as a treatment protocol in our hospital. The present study was conducted to determine the potential benefit of different antiepileptic drugs in reducing agitation among head injury patients admitted to the hospital. There were no significant

**Table 3. Median time in days to achieve Agitation-Relief in head injury patients treated with antiepileptic drugs.**

| Group | n | MS (days) | (CI 95% [LL, UL]) | P value |
|---|---|---|---|---|
| Sod. Valproate | 63 | 3 | 3,3 | 0.0001 |
| Phenytoin | 63 | 4* | 3,4 | |
| Levetiracetam | 63 | 5# | 4,6 | |

MS = Median survival; *p = 0.04; #p = 0.0001 compared to sodium valproate; Peto–Peto–Wilcoxon test was applied.

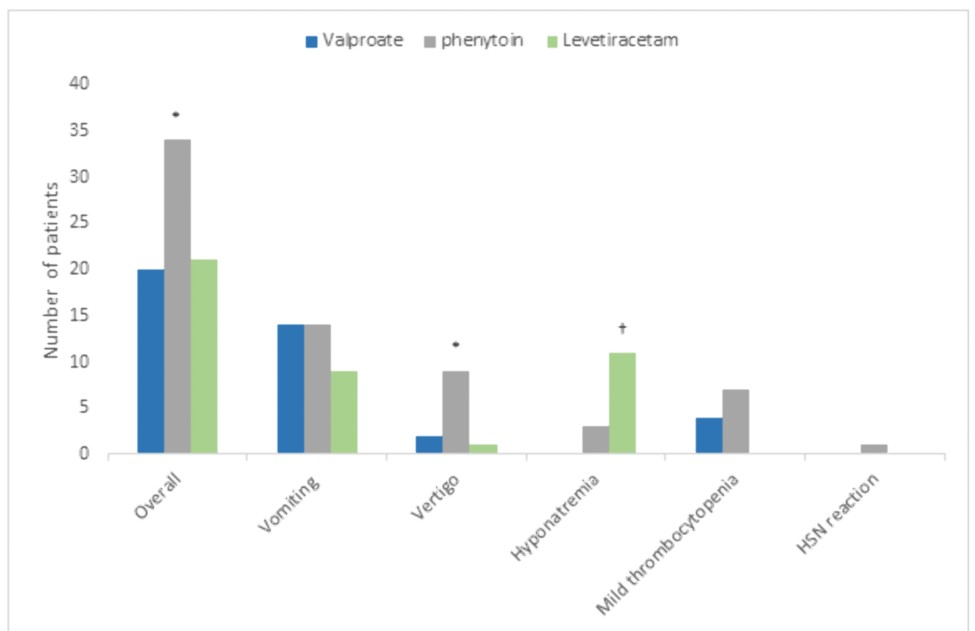

**Fig 3. Pattern of adverse events among groups.** *P < 0.05, phenytoin group compared to other groups. †P < 0.05, levetiracetam group compared to phenytoin. Chi-square test was applied, HSN- hypersensitivity.

differences in the baseline characteristics between the groups, except for a higher percentage of patients in the phenytoin group with a Marshall score grade I compared to sod.v and levetiracetam groups.

Among the three medications used, sod.v was the most effective. Overall, nearly 86% of the patients achieved A-R in the sod.v group compared with phenytoin (75%) and levetiracetam (64%), and this difference was statistically significant (P = 0.016) [Table 2]. Similar to our findings, few studies have shown a downward trend in the prevalence of agitation in head injury patients treated with the prophylactic antiepileptic drug sod.v. [36,37]. A study reported a downward trend in the prevalence of agitation (47.8% to 16.7%) when sod.v was administered for seven days. In another study, the incidence of agitation decreased from 96% to 61% on day 3 following sod.v therapy for agitation in critically ill patients [38–40].

The exact mechanism of beneficial action is not known, but antiepileptic drugs usually suppress CNS excitability. Sodium valproate was found to increase GABAergic signaling via increased release, modulating the signaling of other neurotransmitters, including glutamate, serotonin, and dopamine [41]. It also affects neuronal excitability by blocking voltage-gated sodium and calcium channels and preventing repetitive neuronal firing. In addition to that sod.v upregulates the expression of brain-derived neurotrophic factor (BDNF), which supports neuronal survival and synaptic plasticity,

potentially contributing to its use in mania as a mood stabilizer which has a positive effect on A-R [42]. This multifaceted actions of sodium valproate may account for its greater benefits in agitation compared to other antiepileptic drugs. Phenytoin acts by causing sodium efflux from neurons, which stabilizes the threshold against hyperexcitability [Fig 4]. Levetiracetam is known for its unique mechanism of action primarily through its interaction with the synaptic vesicle protein 2A(SV2A). This interaction is believed to modulate neurotransmitter release, thereby stabilizing neuronal activity, and preventing seizures [43,44].

Phenytoin and levetiracetam are the other antiepileptic drugs used for PTS in our hospital. Our study found A-R rates of 74.6% and 63.5% for phenytoin and levetiracetam groups respectively, although this difference was not statistically significant. There are not enough studies to confirm the role of these drugs in agitation. However, few studies have reported the benefits of phenytoin in controlling aggressive outbursts [45,46]. There is a controversial view regarding the role of levetiracetam in agitation. A few studies have reported that levetiracetam causes agitation, while some have reported that levetiracetam has no significant effect [47,48].

A previous retrospective study reported that agitation was observed in patients treated with levetiracetam, with the median time t first documentation of agitation being 1.3 (0.7–2.7) days after starting the drug [47]. However, in our study, patients with traumatic brain injury treated for more than 4 days were found to experience A-R. The reasons for this disparity may be attributed to the differences in the dose and duration of therapy. Also, because the study was retrospective in nature, its effects were not examined after four days, as the majority of patients presented with agitation within 1.3 days,

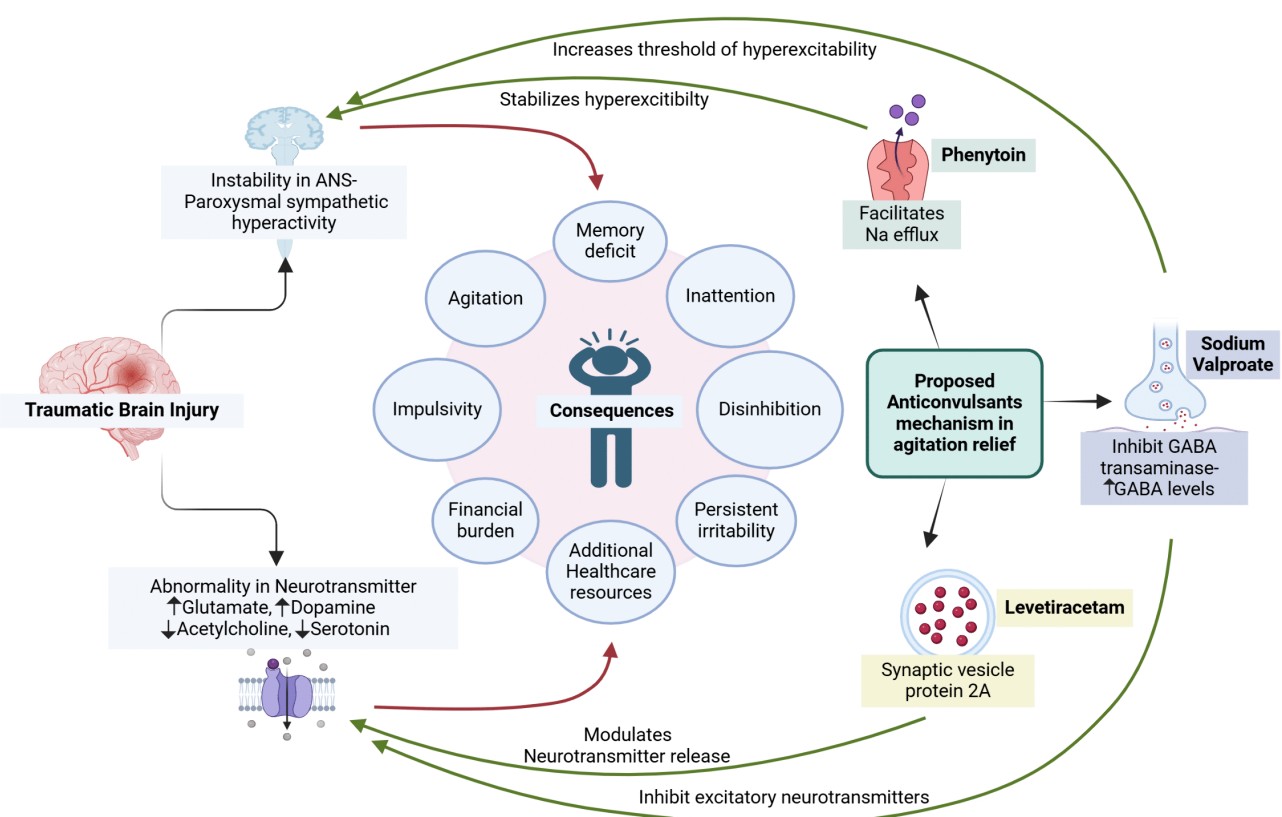

**Fig 4. Potential mechanism of antiepileptic drugs in reducing agitation in traumatic brain.** Created in BioRender. Parida, **A.** (2026) https://BioRender.com/orogd49.

and drug switchover might have occurred early in the study. In our study, the advantage became apparent in the latter half of the study (after four days), and 74% of the participants were found to have experienced A-R by day 6.

Nearly 72% of the patients in the sod.v group experienced A-R within 4 days of therapy. The median time to achieve the event was 3 days in the sod.v group (P = 0.005) compared with 4 days in the phenytoin (P = 0.04) and 5 days in levetiracetam (P = 0.007) groups, indicating early relief from agitation with sod.v drug [Fig 2 and Table 3].

Fewer adverse effects were observed among the patients [Fig 3]. Adverse effects related to drugs included nausea and vomiting, vertigo, hyponatremia, and hypersensitivity. Nausea and vomiting were observed in 39% of the patients in the levetiracetam group and in 30% of those in both the phenytoin and levetiracetam groups. Vertigo was significantly greater in the phenytoin group than in the sod.v and levetiracetam groups. This is a characteristic known adverse effect of phenytoin. Hypersensitivity was observed in one patient who was taking phenytoin, whereas hyponatremia was observed only in the levetiracetam group. Levetiracetam has been reported to cause hyponatremia, similar to our study. The possible mechanisms underlying levetiracetam-induced hyponatremia include Syndrome of Inappropriate Antidiuretic Hormone Secretion (SIADH), altered hypothalamic osmoreceptor sensitivity, and increased renal tubular responsiveness to antidiuretic hormone (ADH) [49–51]. However, liver or renal adverse events were not observed within the duration of the study.

## Limitations

The observational design and sequential allocation during enrollment might have introduced selection bias and further randomized trials is required to validate these findings. As this study focused predominantly on patients with mild traumatic brain injury, exploring the potential benefits in more severe cases would add more information. Additionally, drug blood concentrations were not measured, which may limit the interpretability of the results, particularly for sod.v, given its high plasma protein binding and potential variability in its free fraction. Future studies should focus on elucidating relationship between drug levels and agitation. The findings of this study cannot be broadly generalized, as the sample was limited to hospital-admitted patients, necessitating replication in larger, more diverse populations. While a detailed drug history was obtained, patients who consumed alcohol were excluded on the basis of self-reports without confirmatory blood screening, which could have influenced the results, as agitation trajectories may differ in such patients.

## Conclusion

The study demonstrated that patients receiving sodium valproate showed relatively greater and earlier improvement in agitation control, with a tolerable safety profile. While these findings suggest potential clinical benefit, the treatment selection in post-traumatic brain injury remains individualized, and larger randomized controlled trials are required to determine the optimal therapeutic strategy.

## Author contributions

**Conceptualization:** Raghavendra Nayak, Shivaprakash Gangachannaiah, Varun Kumar S. G., Geeta Sundar.

**Data curation:** Shivani Singh, Raghavendra Nayak, Shivaprakash Gangachannaiah, Reena Sherin Parveen, Varun Kumar S. G.

**Formal analysis:** Reena Sherin Parveen.

**Methodology:** Shivani Singh, Raghavendra Nayak, Shivaprakash Gangachannaiah, Reena Sherin Parveen, Varun Kumar S. G.

**Software:** Shivani Singh, Raghavendra Nayak, Ashutosh Bhosale, Varun Kumar S. G.

**Supervision:** Raghavendra Nayak, Shivaprakash Gangachannaiah, Geeta Sundar.

**Validation:** Shivani Singh, Shivaprakash Gangachannaiah, Ashutosh Bhosale, Geeta Sundar.

**Visualization:** Raghavendra Nayak, Shivaprakash Gangachannaiah, Ashutosh Bhosale, Reena Sherin Parveen, Varun Kumar S. G, Geeta Sundar.

**Writing – original draft:** Shivani Singh, Shivaprakash Gangachannaiah, Geeta Sundar.

**Writing – review & editing:** Raghavendra Nayak, Shivaprakash Gangachannaiah, Ashutosh Bhosale, Reena Sherin Parveen.

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
