## [Decision Letter · Decision Letter 0]

3 Mar 2026

PONE-D-25-68208Comparing the safety and efficacy of sodium valproate, levetiracetam, and phenytoin in attenuating the severity of agitation in patients with post-traumatic brain injury: an observational studyPLOS One

Dear Dr. Gangachannaiah,

Thank you for submitting your manuscript to PLOS ONE. After careful consideration, we feel that it has merit but does not fully meet PLOS ONE’s publication criteria as it currently stands. Therefore, we invite you to submit a revised version of the manuscript that addresses the points raised during the review process.

We look forward to receiving your revised manuscript.

Kind regards,

Giuseppe Biagini, MD

Academic Editor

PLOS One

Journal Requirements:

[NO authors have competing interests].

6. We note that Figure 4 in your submission contains images which may be copyrighted. All PLOS content is published under the Creative Commons Attribution License (CC BY 4.0), which means that the manuscript, images, and Supporting Information files will be freely available online, and any third party is permitted to access, download, copy, distribute, and use these materials in any way, even commercially, with proper attribution. For more information, see our copyright guidelines: http://journals.plos.org/plosone/s/licenses-and-copyright.

1. You may seek permission from the original copyright holder of Figure 4 to publish the content specifically under the CC BY 4.0 license.

Reviewers' comments:

Reviewer's Responses to Questions

**Comments to the Author**

1. Is the manuscript technically sound, and do the data support the conclusions?

Reviewer #1: Yes

2. Has the statistical analysis been performed appropriately and rigorously?

Reviewer #1: No

3. Have the authors made all data underlying the findings in their manuscript fully available?

Reviewer #1: Yes

4. Is the manuscript presented in an intelligible fashion and written in standard English?

Reviewer #1: Yes

5. Review Comments to the Author

Reviewer #1: The authors of PONE-D-25-68208 conducted a prospective study to compare the safety and efficacy of sodium valproate, levetiracetam, and phenytoin in agitation control in patients managed after traumatic brain injury.

The study concludes that sodium valproate has higher efficacy and rapid beneficial actions in reducing agitation compared to levetiracetam, and the authors suggest that this treatment strategy offers better safety profile than phenytoin and levetiracetam.

I believe that the study was well designed and reported, the reference list is extremely poor and would benefit from inclusion of other relevant works. I do not totally agree with the conclusion that sodium valproate is easier to use than levetiracetam and would also modify the conclusion accordingly. Please see my detailed points below:

The introduction would benefit from additional references on delirium incidence, pathophysiology and management in NeuroIntensive Care Unit and in particularly following admission of silver traumas, given the increased risk in elderly patients. Please consider the followings:

- O'Keeffe F, et al. Serum biomarkers of delirium in critical illness: a systematic review of mechanistic and diagnostic evidence. Intensive Care Med Exp. 2025;13(1):90. doi: 10.1186/s40635-025-00795-z.

- Depreitere B, et al. Unique considerations in the assessment and management of traumatic brain injury in older adults. Lancet Neurol. 2025;24(2):152-165. doi: 10.1016/S1474-4422(24)00454-X.

I would also suggest to provide a better rationale for the use of antiepileptics, the authors provide only one paragraph about that. Ganau et al. provided a clear rationale for such pharmacological class and I would suggest to refer to their section dedicated to antiepileptics:

- Ganau M, et al. Delirium and agitation in traumatic brain injury patients: an update on pathological hypotheses and treatment options. Minerva Anestesiol. 2018;84(5):632-640. doi: 10.23736/S0375-9393.18.12294-2.

The antiepileptic drugs suggested are very effective in TBI patients however at present there is no equipoise on the most appropriate strategy, in fact the randomized trial MAST (see https://clinicaltrials.gov/study/NCT04573803) is still enrolling patients to clarify this matter. At present the only consensus is on the use of sedation plus poly-pharmacological strategy for the management of status epileptics following TBI (see Prisco L, et al. A pragmatic approach to intravenous anaesthetics and electroencephalographic endpoints for the treatment of refractory and super-refractory status epilepticus in critical care. Seizure. 2020;75:153-164. doi: 10.1016/j.seizure.2019.09.011.).

A consideration on those important pieces of scientific research is certainly warranted.

6. PLOS authors have the option to publish the peer review history of their article (what does this mean?). If published, this will include your full peer review and any attached files.

Reviewer #1: No

---

## [Author Response · Author response to Decision Letter 1]

7 May 2026

Response: Thank you sir. We have complied with the journal style

[NO authors have competing interests].

Response: We have stated "The authors have declared that no competing interests exist" and the same is included in the Cover letter.

Response: Dear Sir, we have provided the compiled raw dataset used in the study as a supporting information file for your reference along with codes [S1_Rawdata and S2_Diary].

Response: Thank you for the clarification. We confirm that the submission contains all data required to replicate the study findings, including the values underlying summary statistics, figures, and analyses, which are provided as Supporting Information.

The dataset has been fully anonymized and includes all variables necessary for reproducibility. Although informed consent was obtained, explicit consent for unrestricted public data sharing was not secured at the time of enrolment from the participants. We therefore consulted the Institutional Ethics Committee, and in accordance with their guidance, full open access to the dataset is restricted to ensure participant confidentiality. The raw data are available and maintained within the Institutional Ethics Committee repository; access will be provided upon request. Requests for access can be directed to iec.kmc@manipal.edu to ensure compliance with ethical and legal requirements.

Response: Dear Sir, the orcid ID of the corresponding author is validated in Editorial Manager. Corresonding author’s ORCID ID is 0000-0002-6359-4024

6. We note that Figure 4 in your submission contains images which may be copyrighted. All PLOS content is published under the Creative Commons Attribution License (CC BY 4.0), which means that the manuscript, images, and Supporting Information files will be freely available online, and any third party is permitted to access, download, copy, distribute, and use these materials in any way, even commercially, with proper attribution. For more information, see our copyright guidelines: http://journals.plos.org/plosone/s/licenses-and-copyright.

Response: Sir, we thank you for highlighting this important point. Figure 4 was created by myself (Dr Shivaprakash G and Dr Shivani) using BioRender under an institutional academic license.

We have obtained and uploaded the official BioRender “Confirmation of Publication and Licensing Rights – Open Access” document, which explicitly permits publication of the figure under a CC BY 4.0 license, including for journals such as Public Library of Science.

We confirm here that the figure does not contain any third-party copyrighted material. In accordance with BioRender’s licensing requirements, the appropriate citation has been included in the revised figure caption. [Created in BioRender. Parida, A. (2026) https://BioRender.com/orogd49]

We confirm that the figure fully complies with the journal’s copyright and licensing requirements.

Reviewers' comments:

Reviewer's Responses to Questions

Comments to the Author

1. Is the manuscript technically sound, and do the data support the conclusions?

Reviewer #1: Yes

Response: Thank you Sir.

2. Has the statistical analysis been performed appropriately and rigorously?

Reviewer #1: No.

Response: Dear Sir, we thank the reviewer for this important comment. The entire dataset was reanalyzed in consultation with an independent statistician, with analyses performed using the latest version of R software and appropriate statistical methods. All results were cross-verified to ensure accuracy and consistency. The detailed statistical methodology has been clarified in the revised “Statistical Analysis” section, and the raw dataset has been provided for transparency. We confirm that the analyses performed are appropriate and robust for the study and can be independently verified.

3. Have the authors made all data underlying the findings in their manuscript fully available?

Reviewer #1: Yes

Response: Thank you Sir.

4. Is the manuscript presented in an intelligible fashion and written in standard English?

Reviewer #1: Yes

Response: Thank you Sir.

5. Review Comments to the Author

Reviewer #1: The authors of PONE-D-25-68208 conducted a prospective study to compare the safety and efficacy of sodium valproate, levetiracetam, and phenytoin in agitation control in patients managed after traumatic brain injury.

The study concludes that sodium valproate has higher efficacy and rapid beneficial actions in reducing agitation compared to levetiracetam, and the authors suggest that this treatment strategy offers better safety profile than phenytoin and levetiracetam.

I believe that the study was well designed and reported, the reference list is extremely poor and would benefit from inclusion of other relevant works. I do not totally agree with the conclusion that sodium valproate is easier to use than levetiracetam and would also modify the conclusion accordingly. Please see my detailed points below:

The introduction would benefit from additional references on delirium incidence, pathophysiology and management in NeuroIntensive Care Unit and in particularly following admission of silver traumas, given the increased risk in elderly patients. Please consider the followings:

- O'Keeffe F, et al. Serum biomarkers of delirium in critical illness: a systematic review of mechanistic and diagnostic evidence. Intensive Care Med Exp. 2025;13(1):90. doi: 10.1186/s40635-025-00795-z.

- Depreitere B, et al. Unique considerations in the assessment and management of traumatic brain injury in older adults. Lancet Neurol. 2025;24(2):152-165. doi: 10.1016/S1474-4422(24)00454-X.

I would also suggest to provide a better rationale for the use of antiepileptics, the authors provide only one paragraph about that. Ganau et al. provided a clear rationale for such pharmacological class and I would suggest to refer to their section dedicated to antiepileptics:

- Ganau M, et al. Delirium and agitation in traumatic brain injury patients: an update on pathological hypotheses and treatment options. Minerva Anestesiol. 2018;84(5):632-640. doi: 10.23736/S0375-9393.18.12294-2.

The antiepileptic drugs suggested are very effective in TBI patients however at present there is no equipoise on the most appropriate strategy, in fact the randomized trial MAST (see https://clinicaltrials.gov/study/NCT04573803) is still enrolling patients to clarify this matter. At present the only consensus is on the use of sedation plus poly-pharmacological strategy for the management of status epileptics following TBI (see Prisco L, et al. A pragmatic approach to intravenous anaesthetics and electroencephalographic endpoints for the treatment of refractory and super-refractory status epilepticus in critical care. Seizure. 2020;75:153-164. doi: 10.1016/j.seizure.2019.09.011.).

A consideration on those important pieces of scientific research is certainly warranted.

Response: We sincerely thank the reviewer for the constructive feedback and for highlighting relevant literature to strengthen the manuscript.

As suggested, we have substantially revised the Introduction to improve the depth and context of the study. Specifically, we have incorporated additional references addressing the incidence, pathophysiology, and management of delirium in neurocritical care settings, with particular emphasis on older adults and trauma populations.

• Burden: Lines 120-127

• Pathophysiology: Lines 128-136

• Management: Lines 137-165

We have also expanded the rationale for the use of antiepileptic drugs in post-traumatic agitation providing a clearer mechanistic and clinical basis for this pharmacological approach.

• Rationale of using antiepileptics: Lines 167-181

We have revised the conclusion to avoid overstating the comparative ease of use of sodium valproate; the final conclusion is now data-driven and has been modified in accordance with the available evidence.

Conclusion: Lines 395-399

We believe these changes have significantly improved the scientific rigor and contextual relevance of the manuscript and hope will satisfy, point by point to reviewers’ queries.

---

## [Decision Letter · Decision Letter 1]

14 May 2026

Comparing the safety and efficacy of sodium valproate, levetiracetam, and phenytoin in attenuating the severity of agitation in patients with post-traumatic brain injury: an observational study

PONE-D-25-68208R1

Dear Dr. Gangachannaiah,

We’re pleased to inform you that your manuscript has been judged scientifically suitable for publication and will be formally accepted for publication once it meets all outstanding technical requirements.

Kind regards,

Giuseppe Biagini, MD

Academic Editor

PLOS One

Additional Editor Comments (optional):

Reviewers' comments:

Reviewer's Responses to Questions

**Comments to the Author**

1. If the authors have adequately addressed your comments raised in a previous round of review and you feel that this manuscript is now acceptable for publication, you may indicate that here to bypass the “Comments to the Author” section, enter your conflict of interest statement in the “Confidential to Editor” section, and submit your "Accept" recommendation.

Reviewer #1: All comments have been addressed

2. Is the manuscript technically sound, and do the data support the conclusions?

Reviewer #1: Yes

3. Has the statistical analysis been performed appropriately and rigorously?

Reviewer #1: Yes

4. Have the authors made all data underlying the findings in their manuscript fully available?

Reviewer #1: Yes

5. Is the manuscript presented in an intelligible fashion and written in standard English?

Reviewer #1: Yes

6. Review Comments to the Author

Reviewer #1: Authors have conducted a robust revision, their rebuttal letter provides a comprehensive outline of the changes made in the manuscript and I'm pleased to say that they have addressed all reviewers' comments

7. PLOS authors have the option to publish the peer review history of their article (what does this mean?). If published, this will include your full peer review and any attached files.

Reviewer #1: No

---

## [Editor Report · Acceptance letter]

PONE-D-25-68208R1

PLOS One

Dear Dr. Gangachannaiah,

I'm pleased to inform you that your manuscript has been deemed suitable for publication in PLOS One. Congratulations! Your manuscript is now being handed over to our production team.

Kind regards,

on behalf of

Dr. Giuseppe Biagini

Academic Editor

PLOS One